



# Retrieval of UVB aerosol extinction profiles from the ground-based Langley Mobile Ozone Lidar (LMOL) system

Liqiao Lei[1,2], Timothy A. Berkoff[2], Guillaume Gronoff [2,3], Jia Su[4], Amin R. Nehrir[2], Yonghua Wu[5.6], Fred Moshary[5,6], Shi Kuang[7]

[1]Universities Space Research Association, Columbia, MD, USA
[2]NASA Langley Research Center, Hampton, VA, USA
[3]Science Systems and Applications, Hampton, VA, USA
[4]Hampton University, Hampton, VA, USA
[5]City College of New York (CCNY), New York, NY 10031, USA
[6]NOAA – Cooperative Science Center for Earth System Sciences and Remote Sensing Technologies, USA
[7]University of Alabama in Huntsville, Huntsville, Alabama, USA

*Correspondence to*: Liqiao Lei (liqiaolei1007@gmail.com)

**Abstract.** Aerosols emitted from wildfires are becoming one of the main sources of poor air quality in the US mainland. Their extinction in UVB (wavelength range 280-315 nm) is difficult to be retrieved using simple lidar techniques because of the impact of $O_3$ absorption and lacking information of lidar ratio at those wavelengths. The 2018 Long Island Sound Tropospheric Ozone Study (LISTOS) campaign in the New York City region allowed the characterization of lidar ratio for UVB aerosol retrieval. An algorithm for the aerosol extinction retrieval out of the Langley Mobile Ozone Lidar (LMOL) was used in conjunction with the NASA Langley High Altitude Lidar Observatory (HALO) 532 nm aerosol extinction product. This approach requires assuming 2 parameters, the lidar ratio at 292 nm and the Ångström Exponent (AE) between 532 nm and 292 nm. The objective of this work is to determine these two parameters and assess the retrieval error caused by improper assumption of lidar ratio. This work also accomplishes the first know 292nm aerosol product inter-comparison between HALO and Tropospheric Ozone Lidar Network (TOLNet) ozone lidar. HALO results were compared with the aerosol data retrieved from the 292nm band from LMOL with different approximations of the lidar ratio and the AE to determine optimal parameters. Using optimized parameters, the LMOL aerosol extinction can be retrieved with a 10% accuracy up to 3 km. This work highlights the importance of the lidar ratio and AE in the retrieval and validation of 292 nm aerosol profiles obtained from UV-lidar. Errors arise from approaches that utilize a random priori lidar ratio and AE assumption. The lidar ratios at 292 nm determined in this work will also improve our understanding of the UVB optical properties of aerosol in the lower troposphere affected by transported wildfire emission.



# 1 Introduction

Wildfires produce substantial amounts of gaseous pollutants such as carbon monoxide (CO), nitrogen oxides ($NO_x$), volatile organic compounds (VOCs), and ozone ($O_3$) as well as biomass burning particulate which significantly impact the climate and air quality (Andreae and Merlet, 2001; Phuleria et al., 2005; Reid et al., 2005; Zauscher et al., 2013). Pollutants directly emitted from wildfire can affect first responders and local residents. In addition, transported wildfire emission can lead to harmful exposures for populations in regions far away from the wildfires (Cottle et al., 2014; Dreessen et al., 2016). The increase in frequency and severity of North American wildfires significantly affects air quality by increasing the amount of particulates and ozone in the air (Schoennagel et al., 2017). Ground-based lidars have the ability of simultaneously detecting $O_3$ and aerosol with high temporal and vertical resolution to better understand air quality exceedances that can be exacerbated by transported wildfire emission (Aggarwal et al., 2018; Strawbridge et al., 2018; Kuang et al., 2020). On the other hand, the determination of the aerosol properties in UVB wavelength region is of great importance to understand the effect of aerosol on UV radiation which is linked to human health and atmospheric chemistry (Bais et al., 1993; Carlund et al., 2017; Moozhipurath and Skiera, 2020). Lidar aerosol measurements at 355 nm are often reported, but the UVB aerosol properties are rarely studied by lidar (Müller et al., 2007; Nicolae et al., 2013; Haarig et al., 2018). Therefore, we try to retrieval aerosol extinction at 292 nm from the UV-lidar in this work to improve our understanding of the impact of transported wildfire emission on air quality and the aerosol optical properties in UVB band.

Retrieval and validation of lidar aerosol profiles in the UVB wavelengths range are challenging due to 3 factors. First, strong $O_3$ absorption at UVB wavelengths can cause large uncertainty for retrieval of UVB aerosol. One approach to address this is to use $O_3$ measurements to correct the $O_3$ absorption before the extinction/backscatter retrieval technique is applied (Browell 1985; Young, 1995). In the second factor, the information of the lidar ratio at UVB wavelengths is not well established for different aerosol types. In general, a relative lidar ratio is needed to retrieve an accurate UVB aerosol profile. For example, Kuang et al. (2020) demonstrated retrieval of aerosol 299 nm backscatter from the ozone lidar raw attenuated backscatter signal using an iteration algorithm and fixed lidar ratio (60 sr) in the presence of smoke. However, it will introduce uncertainty for the aerosol retrieval if we use one lidar ratio value for aerosol with different type because the lidar ratio for different aerosol type varies a lot (Omar et al., 2009; Lopes et al., 2013, Müller et al., 2007; Burton et al., 2014; Haarig et al., 2018). An accurate lidar ratio at UVB wavelength range is needed to obtain the best accuracy for UVB aerosol profile retrievals. The third factor is the lack of available aerosol profiles at UVB wavelength range to validate the retrieved aerosol result. However, aerosol profiles provided by a more common 532nm/355nm aerosol lidar could be used for validation if the AE between the UVB wavelength and 532nm/355nm is available. This work will focus on addressing these 3 factors and retrieve aerosol extinction at 292 nm for Lnalgye Mobile Ozone Lidar (LMOL) system.

In this work, the 292 nm elastic signal from LMOL is corrected for ozone absorption, and then the Fernald method was applied to retrieve aerosol extinction at 292 nm (Fernald et al., 1972, Fernald, 1984). In this paper, the impact of the aerosols was low enough that an aerosol correction to the $O_3$ density was not necessary; otherwise, an interative process would have been



necessary (Browell et al., 1985). Since an inaccurate lidar ratio value can cause an error in the profile retrieval, we propose a method to determine the lidar ratio at 292 nm using partial aerosol optical depth (AOD) difference. The Long Island Sound Tropospheric Ozone Study (LISTOS) campaign is a multi-agency collaborative study for the areas of Long Island Sound and surrounding coastlines. The LMOL and the airborne HALO system were both operating during the LISTOS campaign in 2018 summer. As figure s1 shows, the HALO and LMOL will have coincident measurement when the HALO overpass the LMOL

site. The coincident HALO aerosol profile and the LMOL measurement provide us opportunity to determine the lidar ratio at 292 nm to improve the LMOL UVB aerosol retrieval. The proper AE between the 532 nm and the 292nm also was selected simultaneously. The information about lidar ratio at 292 nm and AE between 292 nm and 532 nm not only improve our UV aerosol retrieval and validation, but also improve our understanding of the aerosol optical properties at those wavelengths. During the LISTOS campaign, the case study on August 2018 were selected as an example of the UVB aerosol retrieval

because HALO has coincident aerosol data with LMOL data that provides us the opportunity to do desired analysis. The August 28 case was shown in detail because the air quality exceedance during that day was probably caused by the impact of long-range transport of wildfire emissions (Rogers et al., 2020). It is important to understand the aerosol optical properties for this air quality exceedance case.

In this paper, instruments and data used in this work are introduced in the next section. We use the LMOL raw and $O_3$ data

products, HALO aerosol backscatter/extinction and lidar ratio data, the City College of New York (CCNY) 532 nm aerosol extinction data, and the CL51 backscatter data. The method to retrieve UVB aerosol extinction, as well as the method to select the lidar ratio for UVB aerosol retrieval is presented in the following section. The comparison between the retrieved LMOL aerosol extinction profile and the HALO aerosol extinction profile are presented and compared with CL51 and CCNY aerosol lidar data. Finally, the uncertainty for the aerosol extinction retrieval is analysed before final discussion and conclusions.

## 85 2 Instrument and data

### 2.1 The LMOL system

LMOL is a mobile ground-based $O_3$ differential absorption lidar (DIAL) system that has a transmitter with a 1 kHz diode-pumped Q-switched Nd:YLF 527 nm laser to pump a custom-built Ce:LiCAF tunable UV laser to generate "on" and "off" DIAL wavelengths at 286 nm and 292 nm. A 40 cm diameter telescope was used to collect the back scatter signal for the far-

field and a smaller diameter wide field off-axis parabolic mirror is used for the near-field return (De Young et al., 2017; Farris et al., 2019). Both far-field and near-field receiver channels employ analog and photon detection modes using a high-speed Licel data acquisition system to maximize measurement dynamic range. The current configuration of LMOL can retrieve $O_3$ profiles from 0.1 to 10 km range at night, with 5 to 10-minute temporal averaging (Gronoff et al. 2019, 2021, Farris et al., 2019). During daytime, the maximum altitude reached is typically close to 5 km due to solar background light limitations.

LMOL is part of Tropospheric Ozone Lidar Network (TOLNet) (https://www-air.larc.nasa.gov/missions/TOLNet/), a network of $O_3$ lidars that help evaluate air quality models and compliment current and planned satellite retrievals for satellite such as





the Tropospheric Emissions: Monitoring of Pollution (TEMPO) mission (Zoogman et al., 2017). LMOL generates data products following the TOLNet protocol for the acquisition, processing, and archiving of the data that assure the quality and consistency of the data products (Leblanc et al., 2016a; Leblanc et al., 2016b; Leblanc et al.,2018). For LMOL data products,

the vertical resolution (110 m to 990m) of the $O_3$ profiles varies with altitude to preserve a retrieval uncertainty within $\pm 10\%$, the uncertainty of which is calculated using poison statistics of the backscattered photons. LMOL has been used in several campaigns such as Ozone Water-Land Environmental Transition Study (OWLETS) I and II, LISTOS (*Berkoff et al.*,2018; *Sullivan et al.,* 2019; *Dacic et al.*, 2020), Fire Influence on Regional to Global Environments and Air Quality (FIREX-AQ), and Southern California Ozone Observation Project (SCOOP) (Leblanc et al., 2018). In the context of LISTOS (Wu et al.,

2021), and more specifically for the present study, LMOL was deployed at Sherwood Island Park, Westport, CT (41.1182° N, 73.3368° W, 2.5 m ASL) and obtained measurements between July 12 and August 29, 2018. To obtain the aerosol products, we used the LMOL raw data at 292nm and the LMOL $O_3$ data.

**2.2 The HALO aerosol measurement**

The NASA airborne High Altitude Lidar Observatory (HALO) is a combied High Spectral Resolution Lidar (HSRL) and $H_2O$

and $CH_4$-differential absorption lidar (DIAL) (Nehrir et al., 2017; Wu et al., 2021). HALO employs 1 KHz Nd:YAG pumped optical parametric oscillators to generate the DIAL wavelength for $H_2O$ and $CH_4$ observations. The residual energy from the conversion process is employed for the HSRL technique. HALO employs the HSRL technique at 532 nm, the backscatter technique at 1064nm, and measures depolarization at both 532/1064 nm. And I2 vapor cell is used to in the receiver to separate the molecular scattering from the total scattering (Hair et al., 2008). This allows for discrimination of aerosol scattering from

molecular and retrieval of aerosol extinction and backscatter coefficient independently (Burton et al., 2013, 2014, 2015, Hair et al, 2008). The lidar extinction-to-backscatter ratio is then available from the HALO determined aerosol extinction and backscatter coefficients. HALO data are sampled at 0.5-s temporal and 1.25 m vertical resolution. This vertical resolution for the aerosol measurement is increased to 15 m in post-processing to increase the SNR of the aerosol intensive and extensive retrievals. Aerosol backscatter and depolarization products are averged 10 s horizontally and aerosol extinction products are

averaged 60 s horizontally and 150 m vertically. The polarization and HSRL gain ratios are calculated as described in Hair et al., 2008. Operational retrievals also provide mixing ratio of non-spherical -to-spherical backscatter (Sugimoto and Lee, 2006), aerosol type, (Burton et al., 2012) and aerosol mixed layer height (Scarino et al., 2014). In this study, the HALO aerosol extinction data are selected when its flight measurements are overpass the LMOL site.

**2.3 The CCNY aerosol lidar**

The CCNY lidar transmits 1064, 532, and 355 nm with a flash lamp-pumped Nd: YAG laser with a pulse repetition rate of 30 Hz. A telescope with 50 cm diameter collects three-wavelength elastic scatter and two Raman-scattering returns (by nitrogen and water vapor excited by 355 nm laser). The aerosol extinction and backscatter profiles in the troposphere were retrieved





and the AE was derived to distinguish fine mode aerosol from coarse mode aerosol. CCNY lidar return signals detection start from 0.5 km with a 1-min time average and 3.75 m vertical data-bin resolution. The PBL height was estimated from the 1064

nm elastic return because the backscatter signal in this wavelength is more sensitive to aerosol structures than shorter wavelength (Wu et al., 2019; Wu et al., 2018, Wu et al. 2021). CCNY lidar was located at New York City (NYC) (40.8198° N, -73.9483° W) to remote sensing the aerosol layer aloft during the LSITOS campaign.

**2.4 The Ceilometer located nearby LMOL**

A ceilometer (Vaisala CL51) was installed at the Westport site co-located with LMOL at 41.1173N, 73.3369 W, 3 m above

sea level during the LISTOS campaign. A ceilometer is a single-wavelength backscatter lidar system used to monitor cloud base height and aerosol structures (Wang et al., 2018). A semiconductor laser (InGaAs diode laser) with 3.0 uJ pulse energy and repetition rate of 6.5 kHz retrieves the atmospheric backscatter at 910 nm to infer the vertical distribution of clouds and aerosols up to 15 km (Lee et al., 2018; Jin et al., 2015). The measured backscatter signal was integrated over 5 seconds. It is an autonomous eye-safe system which obtains measurements makes 24-hr/7-day observations. Although the molecular signal

returns are weak because of the low-energy laser and the near-infrared wavelength, the stronger returns from aerosols and clouds can be detected.  The CL51 signal is impacted by dark current noise and daytime solar background, but still sufficient to measure signals from boundary layer aerosols up to 3 km (Jin et al., 2015). As a result, the ceilometer can provide the boundary layer evolution and aerosol retrievals up to 3 km to qualitatively compare with LMOL.

**3 Methodology**

**3.1 Method to retrieve aerosol extinction coefficient**

LMOL uses the 287 nm and 292 nm wavelengths for $O_3$ DIAL measurements. The 292 nm "off" wavelength was selected for the aerosol retrieval in this work because $O_3$ has a smaller absorption cross section at this wavelength. The attenuated lidar signal measured by the LMOL system can be represented by,

$$P_\lambda(R) = \frac{C_\lambda\left(\beta_{\lambda,a}(R)+\beta_{\lambda,m}(R)\right)\left\{\exp\left[-2\int_0^R\left(\alpha_{\lambda,a}(r)+\alpha_{\lambda,m}(r)+\sigma_{\lambda,O_3}N_{O_3}(r)\right)dr\right]\right\}}{R^2} + P_0 \qquad (1)$$


where $P_\lambda(R)$ is lidar return signal power, $\lambda$ is laser wavelength, $C_\lambda$ is lidar system constant, $\beta_{\lambda,a}(R)$ is aerosol volume backscatter coefficient, $\beta_{\lambda,m}(R)$ is molecular volume backscatter coefficient, $\alpha_{\lambda,a}(R)$ is aerosol optical extinction coefficient, $\alpha_{\lambda,m}(R)$ is molecular optical extinction coefficient (without the $O_3$ extinction), $\sigma_{\lambda,O_3}$ is the $O_3$ absorption cross section, $N_{O_3}(R)$ is the $O_3$ number density. $P_{\lambda,0}$ is the offset which contributed by the sky background signal, amplifier and digitizer offset, and

detector dark current (Fernald, 1984; Young et al., 2009). We also have $\alpha_{\lambda,m}(R) = \sigma_m N_m$ where $\sigma_m$ is the atmospheric extinction cross section and $N_m$ is the atmospheric molecular number density. The molecular extinction coefficient and



backscatter coefficient are usually calculated from the balloon measurement close to the lidar site or from model like GEOS 5 (Sasano and Nakane, 1984).

We denoted the aerosol extinction-to-backscattering ratio (also known as lidar ratio) as $S_1 = \alpha_{\lambda,a}/\beta_{\lambda,a}$ , and the molecular

extinction-to-backscatter ratio as $S_2 = \alpha_{\lambda,m}/\beta_{\lambda,m} = 8\pi/3$ (Kovalev and Eichinger, 2004). We have assumed a constant $S_1$ with range for the aerosol extinction retrieval (Fernald, 1972; Fernald, 1984). The received LMOL lidar signal at 292 nm could be corrected with the ozone profile to get the elastic lidar attenuated backscatter signal attributed to aerosol and molecular terms as shown in equation (2). The $O_3$ corrected range-corrected lidar signal with background subtraction is shown as following:

$$[P_\lambda(R) - P_{\lambda,0}]R^2\{\exp\left[2\int_0^R \sigma_{\lambda,O_3}N_{O_3}(r)\,dr\right]\} = C_\lambda\left(\beta_{\lambda,a}(R) + \beta_{\lambda,m}(R)\right)\{\exp\left[-2\int_0^R\left(\alpha_{\lambda,a}(r) + \alpha_{\lambda,m}(r)\right)dr\right]\} \qquad (2)$$

We can rearrange Equation (2) to get the aerosol attenuated backscatter signal X(R):

$$X(R) = C_\lambda\left(\beta_{\lambda,a}(R) + \beta_{\lambda,m}(R)\right)\{\exp\left[-2\int_0^R\left(\alpha_{\lambda,a}(r) + \alpha_{\lambda,m}(r)\right)dr\right] \qquad (3)$$

where $X(R) = [P_\lambda(R) - P_0]R^2\{\exp\left[2\int_0^R \sigma_{\lambda,O_3}N_{O_3}(r)\,dr\right]\}$. Use equation (3) and the aerosol and molecular extinction-to-

backscattering ratio, the aerosol extinction coefficient at ranges between the lidar and calibration range $R_c$ is shown in equation (4) (Fernald, 1984; Sasano et al., 1985). The equations after here do not include $\lambda$ for convenience.

$$\alpha_a(R) + \frac{S_1}{S_2}\,\alpha_m(R) \quad = \frac{X(R)\{\exp\left[-2\left(\frac{S_1}{S_2}-1\right)\int_{R_c}^R \alpha_m(r)dr\right]\}}{\frac{X(R_c)}{\alpha_a(R_c)+\frac{S_1}{S_2}\alpha_m(R_c)}-2\int_{R_c}^R X(r)\{\exp\left[-2\left(\frac{S_1}{S_2}-1\right)\int_{R_c}^r \alpha_m(r')dr'\right]\}dr} \qquad (4)$$

In order to calculate the aerosol extinction coefficient $\alpha_a(R)$, we need to assume an aerosol lidar ratio $S_1$ and the reference value of the aerosol extinction coefficient at a calibration range $R_c$. The reference value $\alpha_a(R_c)$ must be known or

estimated. The calibration range and the reference value could be estimated use the secant method mentioned by Li et al, 2018. We need pay attention to that all data used in the aerosol retrieval process should has same vertical resolution. The retrieval is applied to cloud-free profiles after applying a cloud screening on the data. This was done by using the convolution of the $O_3$ corrected attenuated backscatter signal and a Harr wavelet function to identify cloud edges and then further screened by using a threshold to separate cloud features (Burton et al., 2010, Compton et al., 2013; Scarino et al., 2014). The aerosol extinction

was retrieved for both LMOL far-field-photon-counting and far-field-analog signal channels. The near field aerosol retrieval will be described in a separate work. The aerosol extinction profiles for those two channels were merged to a single profile with overlapping altitude zone 1.5-2 km. The lowest altitude for the retrieved profile is about 0.5 km with the highest altitude for retrieved aerosol being constrained by the highest altitude of reliable $O_3$ data.






## 3.2 Selection of the UVB $S_1$ for retrieval

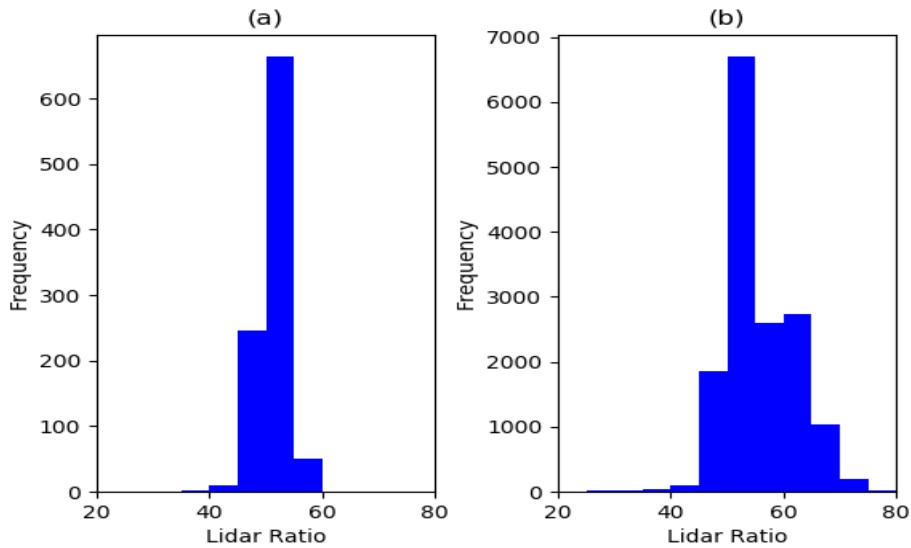

Figure 1: The histogram of average HALO $S_1$ frequency distribution for (a) August 28fternoon, and (b) August 5, 6, 15, 16, 24, 28, and 29, 2018 measurement during LISTOS campaign.

The O₃ corrected LMOL attenuated backscatter profile does not contain information needed to estimate $S_1$. The $S_1$ value is dependent on the particle size, shape, and refractive index, and usually varies from ~10 to 100 sr (Sasano and Nakane, 1984). Fortunately, the HALO observations provide the 532 nm $S_1$ which could help us learn some information about the aerosol optical properties for the cases where the two instruments have coincident observations. As mentioned in section 1, we focus on case studies from August 2018, especially during the afternoon period of August 28, 2018. The average $S_1$ for HALO at

532 nm $S_1$ profiles was calculated for August 28, 2018, afternoon data. The $S_1$ mentioned hereafter are the vertically average $S_1$ derived from HALO $S_1$ profile. The frequency distribution of the HALO $S_1$ for the afternoon August 28, 2018, is shown in Figure 1 (a). The mean HALO $S_1$ for 532 nm is ~ 55 sr with a 1-σ standard deviation ~3 sr. As figure 1 (b) show, the mean HALO $S_1$ for all available August measurement is ~55 with 1-σ  standard deviation ~6 sr. The HALO 532 nm $S_1$ data was screened by criteria of $S_1$ larger than 10 and less than 100 sr when calculating the average for each $S_1$ profile.

The following paragraph will introduce the method to identify the $S_1$ at 292 nm and extinction AE between 292nm and 532nm by calculating the partial AOD difference between the retrieved LMOL 292 nm aerosol extinction profile and HALO aerosol extinction profile. The AE represents the wavelength dependency of the AOD or extinction coefficient for aerosol. The AE (noted as $\alpha_{\lambda_1,\lambda_2}$) between two wavelengths $\lambda_1$ and $\lambda_2$ is expressed as the following equation (Wagner and Silva, 2008):

$$\alpha_{\lambda_1,\lambda_2} = -\frac{\ln\left(\frac{\tau_1}{\tau_2}\right)}{\ln\left(\frac{\lambda_1}{\lambda_2}\right)} \tag{5}$$





where $\tau_1$ and $\tau_2$ are the AOD at wavelength $\lambda_1$ and $\lambda_2$.

The ideal $S_1$ at 292 nm and AE between 292 nm and 532 nm were determined from the retrieved LMOL aerosol extinction profiles and aerosol profiles provided by co-located HALO measurements using a partial AOD difference method. Figure 2 shows the flow chart for 5 steps of this partial AOD difference method. In step 1, the LMOL aerosol extinction was retrieved by incrementing $S_1(i)$ from 10 sr to 90 sr in steps of 5 sr and notes as $\alpha_{LMOL,S_1(i)}(R)$. Step 2, The LMOL aerosol extinction

$\alpha_{LMOL,S_1(i)}(R)$ multiply $\Delta R$ (7m in this work) to get the LMOL partial AOD at altitude R which noted as $PAOD_{LMOL,S_1(i)}(R)$. Step 3, HALO 532 nm aerosol extinction was converted to aerosol extinction at 292 nm with AE (j) and varied from 0.5 to 2.5 with step 0.1. Then the 292nm HALO aerosol extinction $\alpha_{HALO,AE(j)}(R)$ is multiplied by $\Delta R$ to obtain the HALO partial AOD at altitude R which was noted as $PAOD_{HALO,AE(j)}(R)$. Step 4, the relative difference of partial AOD (noted as $\Delta PAOD_{i,j}(R)$) between $PAOD_{LMOL,S_1(i)}(R)$ and $PAOD_{HALO,AE(j)}(R)$ was calculated using equation (6). Step 5, the $\Delta PAOD_{i,j}(R)$ were

integrated to get the partial AOD difference index $PAODI(i,j)$ using equation (7). In equation (7), $R_b$ and $R_t$ is the bottom and top altitude for calculating the $PAODI(i,j)$. The $S_1(i_c)$ and $AE(j_c)$ correspond to the $S_1$ and $AE$ (532nm & 292 nm) from which we obtain minimums PAODI ($PAODI_{min}$). $S_1(i_c)$ and $AE(j_c)$ is taken as the value that should be used in LMOL aerosol retrieval and HALO aerosol comparison.

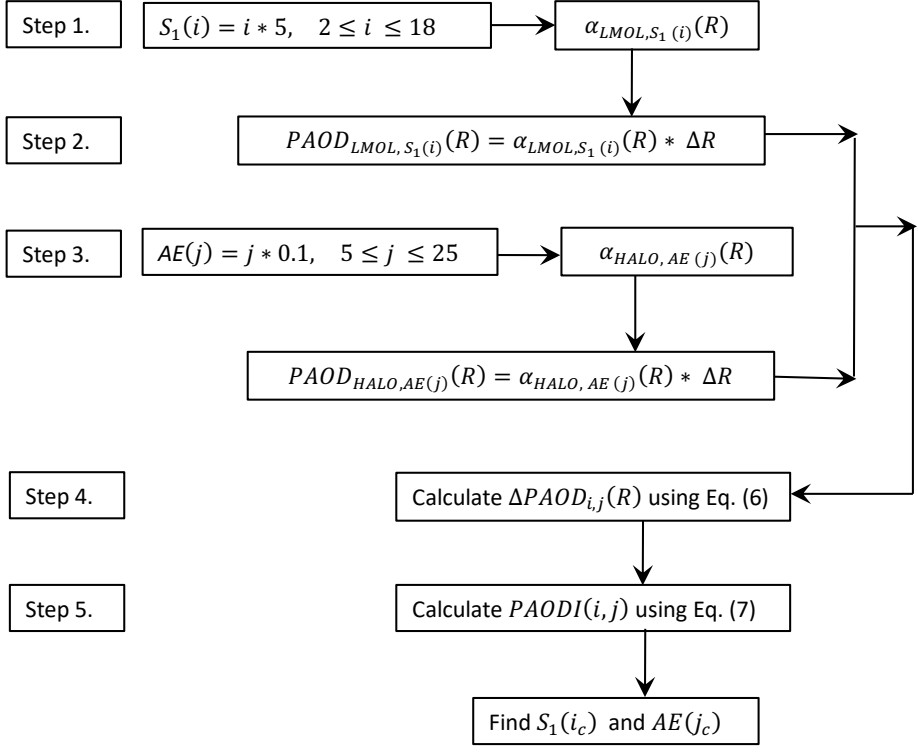

Figure 2: The flow chart for process of calculating the 292 nm $S_1$ and AE between 532 nm and 292 nm. "i" is the integer increment from 2 to 18 that used to calculate the $S_1$ to make the $S_1$ varies from 10 to 90. "j" is the integer increment from 5 to 25 that used to calculate the AE to make the AE varies from 0.5 to 2.5. "I" and "j" also could be the index of the calculated partial AOD.





$$\Delta PAOD_{i,j}(R) = \frac{abs\left[PAOD_{LMOL,S_1(i)}(R) - PAOD_{HALO,AE(j)}(R)\right]}{\left[(PAOD_{LMOL,S_1(i)}(R) + PAOD_{HALO,AE(j)}(R))/2\right]} \tag{6}$$

$$PAODI(i,j) = \sum_{R=R_b}^{R_t} \Delta PAOD_{i,j}(R) \tag{7}$$

In order to show how the PAODI changed with the 292 nm $S_1$ and the AE (292 & 532 nm), we further calculate the percentage relative difference of the PAODI compared with $PAODI_{min}$. An example of this partial AOD difference method at 13:17 EDT on August 28, 2018, was shown in figure 3. The PAODI was calculated for altitude regions from 0.5 to 3 km. The result in figure 3 (a) show that the selected $S_1$ is 35 sr and selected AE (292 & 532 nm) is 1.4. Therefore, the $S_1$ of 35 sr is the ideal choice for aerosol extinction retrieval on August 28, 2018, afternoon. As show in figure 3 (b) the LMOL $S_1$, and AE (292 & 532 nm) at (40, 1.5) and (30,1.3) also has PAODI value very close to PAODImin and could be potential choice for the LMOL retrieval and comparison. Significant errors can arise when improper $S_1$ is used for any UV aerosol retrieval that requires an inversion. For example, the value of PAODI using S₁ = 60 and AE = 1.4 is about 200% of PAODI value using S₁ = 35 and AE = 1.4. The furtherer the S₁ deviates goes away from the correct value, the larger the error will be caused for the UVB aerosol retrieval.

$$PAODI_{rel-diff}(i,j) = \frac{PAODI(i,j) - PAODI_{min}}{PAODI_{min}} \times 100\%. \tag{8}$$

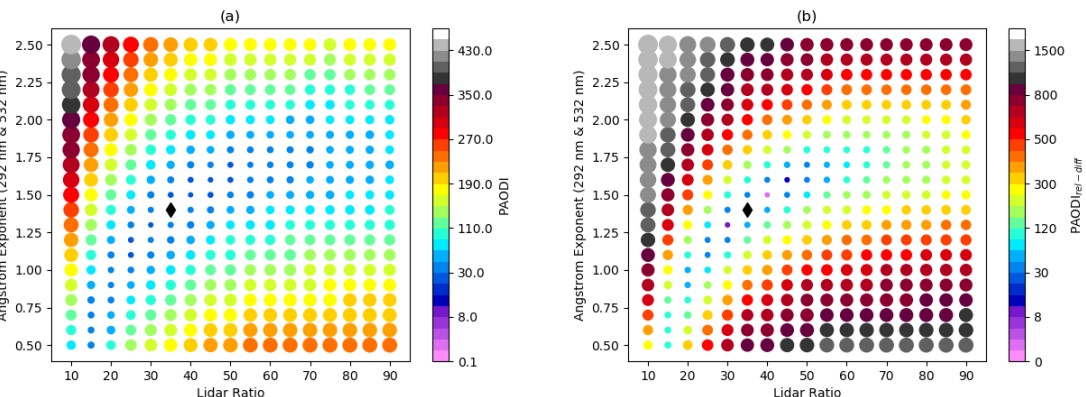

Figure 3: (a) The distribution of PAODI according to the 292 nm $S_1$ and AE (292 nm & 532 nm). Marker color and size show the value of PAODI. The $PAODI_{min}$ was found and the corresponding $S_1$ and AE (292 nm & 532 nm) are noted as black diamond. (b) The distribution of $PAODI_{rel-diff}$. Marker size and color show the value of $PAODI_{rel-diff}$. Black diamond shows the minimum value of $PAODI_{rel-diff}$ and represents the ideal point used for optimized 292 nm aerosol retrieval.

The selected 292 nm S₁ and AE (292 & 532 nm) were derived from the process mentioned above from all available co-located HALO and LMOL measurements for August 5, 6, 16, 24, 28, and 29 The results and the HALO average $S_1$ at 532 nm are





shown in Table 1. The altitude range for calculating the 292 nm $S_1$ and AE (292 & 532 nm) are 0.5 to 3 km with exception of

the afternoon Aug 6 flight, morning flight of Aug 16, and afternoon flight Aug. 29.  In these cases, the altitude range from 0.5

to 2.5 km were used to avoid cloud interferences that prevented proper retrieval and are marked by star on the AM/PM in

Table 1.

Table 1: The LMOL $S_{1,292}$, HALO $S_{1,532}$ and the HALO AE (292 & 521 nm) for August 2018

| Date | Aug. 05 | | Aug. 06 | | Aug. 16 | | Aug. 24 | | Aug. 28 | | Aug. 29 | |
|---|---|---|---|---|---|---|---|---|---|---|---|---|
| | AM | PM | AM | PM* | AM* | PM | AM | PM | AM | PM | AM | PM* |
| AE | \ | 1.7 | 1.2 | 1 | 1 | 1.1 | 1.4 | 1.5 | 1.6 | 1.4 | 1.6 | 1.4 |
| $S_{1,292}$ | \ | 45 | 20 | 20 | 30 | 55 | 40 | 25 | 35 | 35 | 25 | 50 |
| $S_{1,532}$ | \ | 48.6 | 52.1 | 53.9 | 62.7 | 66.7 | 46.7 | 49.4 | 51.7 | 52.1 | 46.8 | 55.5 |

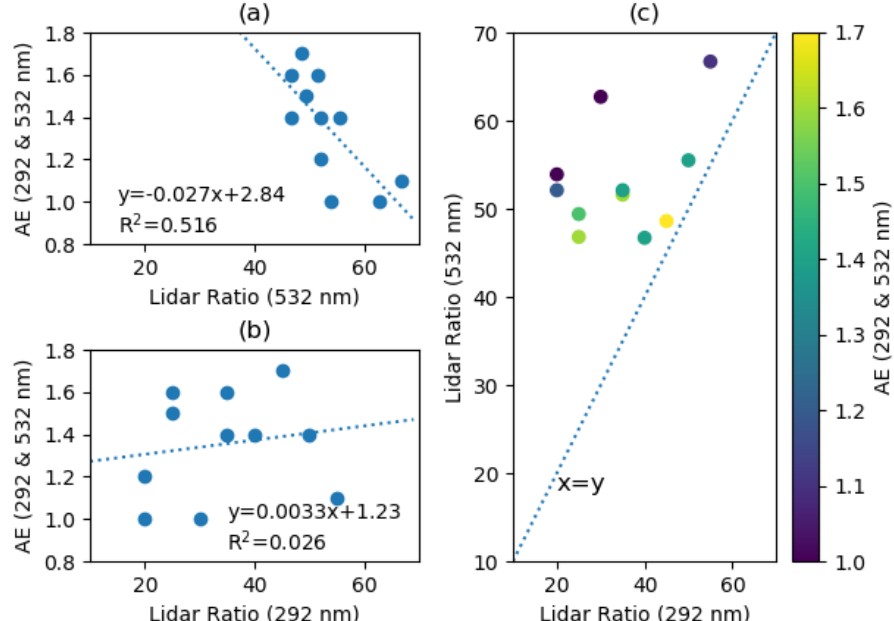

Figure 4: The $S_1$ for 292 and 532 nm, and the AE (292 & 532 nm) according to Table 1. (a) 532 nm $S_1$ and AE (292 & 532 nm) (b) 292 nm $S_1$ and AE (292 & 532) (c) scatter plot of the $S_1$ for 532 nm and 292 nm with color show the AE (292 & 532 nm).

Figure 4 shows the $S_1$ and AE for 292 and 532 nm providing a view of the relationships. As shown in Figure 4 (a) and (b), 532

nm $S_1$ varied between 40 sr and 70 sr and 292 nm $S_1$ varied between 20 sr and 55 sr with AE (292 & 532 nm) varied from 1

to 1.7. Also, it shows in figure 4 (a) that the 532nm $S_1$ are anti-correlated with AE (532 & 292 nm) with correlation coefficient

= -0.72 and R square = 0.516. The anti-correlation indicates that the $S_1$ values dependent on the particle size (Giannakaki et

al., 2010). The 292 nm $S_1$ does not have a clear correlation with AE (532 & 292nm) which is probably caused by the different





aerosol absorption characteristic at 292 nm. Figure 4 (c) shows that 292 nm lidar ratio smaller than 532 nm lidar ratio for all cases listed in Table 1. The smaller lidar ratio at UV wavelength compared with that in visible 532 nm shows the characteristic feature of aged smoke particle (Wandinger et al., 2002; Haarig et al., 2018, Müller et al., 2005; Müller et al., 2007; Ortiz-

Amezcua., 2017). This confirms the previous reports that the air parcel arriving northeastern US has passed over active fires in the southeastern US, northwestern US or Columbia British region (Wu et al., 2021; Rogers et al., 2020; Hung et al., 2020).

## 4 Result

### 4.1 Comparison of retrieved LMOL Result and HALO aerosol extinction profile

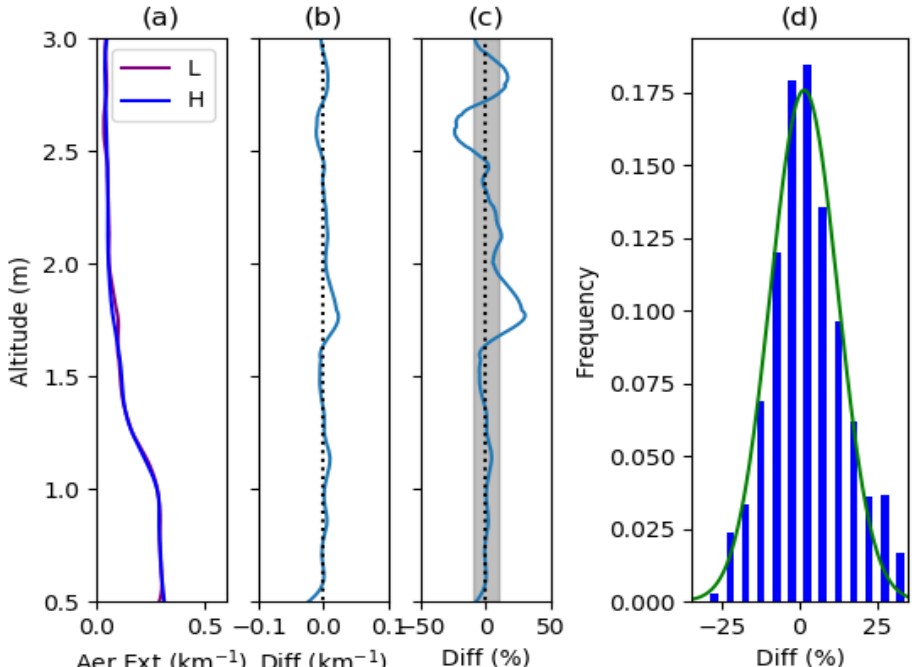

Figure 5: Comparison of the LMOL and HALO derived 292 nm aerosol extinction coefficient on August 28, 2018, afternoon at 13:17 EDT using the $S_1$ and AE selected in section 3.2. The HALO aerosol extinction profile is converted from 532 nm aerosol extinction product. (a) LMOL and HALO aerosol extinction; (b) difference between LMOL and HALO aerosol extinction; (c) The percent difference between LMOL and HALO aerosol extinction; and (d) The error (percentage difference) probability distribution function for all available comparison between 0.5-2.5 km for August 2018. The width between each bar shows 5% difference.

The optimized 292 nm $S_1$ and AE (292 & 532 nm) selected in table 1 was used to retrieve the LMOL 292 nm aerosol extinction and convert the 532 nm HALO aerosol profile to 292 nm to evaluate the retrieved 292 nm aerosol profile. The result of inter-comparison between the retrieved LMOL 292 aerosol extinction and the converted HALO 292 aerosol extinction is shown in figure 5 (a)-(c) for afternoon August 28, 2018. In figure 5 (a), the LMOL 292 nm aerosol extinction profile was shown in





purple and HALO aerosol extinction profile was shown in blue. As shown in figure 5 (c), the percent difference is typically

less than 10 % between 0.5 and 3 km. The grey shadow region in the figure 5 (c) show the ±10% region. The percentage

difference is larger at higher altitudes because the aerosol concentration is lower above the boundary layer resulting in a larger

percentage difference. The percentage difference for all available HALO and LMOL aerosol data between 0.5 to 2.5 km was

used to calculate the probability distribution function of the percentage difference for 5% binning. The result in figure 5 (d)

shows that the distribution of the frequency is centered about zero and exhibited by a gaussian distribution. The total number

of points used for the comparison is 3146. The height of the peak of the distribution function is 0.175 (since it is normalized

to 1). The median error percentage is 1.5% with a standard deviation of 11%. These results show that by using the selected $S_1$

and AE in Table 1, LMOL has the capability to retrieve aerosol extinction in 292nm with reasonable accuracy. This result also

provides the aerosol extinction value in the UVB wavelength range which helps to understanding of the UV aerosol optical

properties transported wildfire smoke aerosol.

**4.2 Comparisons between LMOL, CCNY lidar, and CL51**

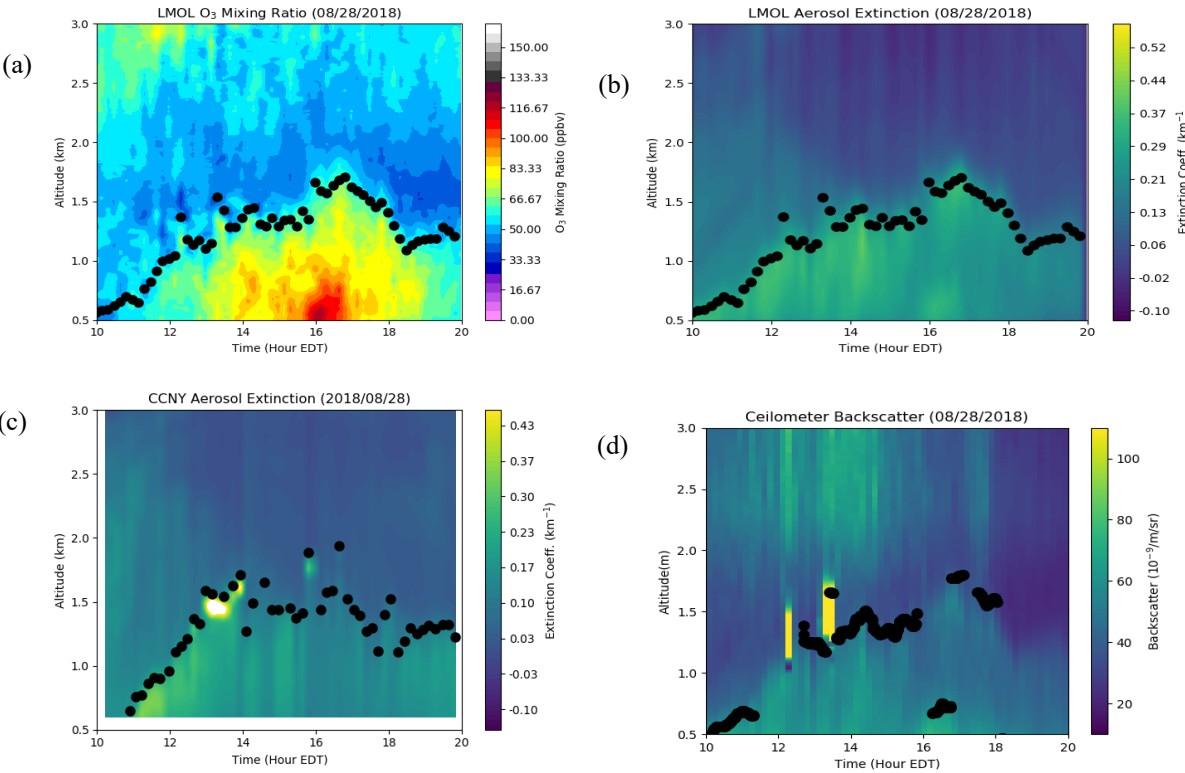

Figure 6. (a) The $O_3$ variation on August 28, 2018 (b) Retrieved LMOL UVB aerosol extinction coefficient curtain plot on August 28,
2018 (c) Same day CCNY lidar aerosol extinction coefficient (converted to 292 nm). (d) Same day 910nm ceilometer CL51 (same
location as LMOL) backscatter. The black dot on the curtain plot of (b), (c), and (d) show the planetary boundary layer (PBL) height.



To examine the LMOL retrieval beyond those times limited to just the HALO overpasses, the August 28 curtain plot of the LMOL 292 nm aerosol extinction also compared with a co-located Ceilometer CL51 backscatter signal, and the CCNY lidar (located in NYC) aerosol extinction. This allowed us observe boundary layer development and examine the aerosol variation

features during the course of the day. The aerosol mixed layer height increases after 10 am EDT and reaches a maximum at 17 EDT. The comparison between the CCNY aerosol extinction and the LMOL aerosol extinction shows that retrieved LMOL UV aerosol extinction are quantitatively consistent. The difference of the aerosol extinction between the LMOL and CCNY measurement probably caused by the atmosphere variation in different locations with about 60 km distance. Planetary boundary layer (PBL) height was retrieved by applying a wavelet method to the LMOL and CCNY aerosol data (Brooks et

al., 2003; Compton et al., 2013; Scarino et al., 2014). PBL height of the ceilometer was obtained from the CL51 aerosol data product. PBL height were overplotted on the aerosol and $O_3$ curtain plot in figure 6 (a) - (d). LMOL retrieved UVB aerosol extinction, and co-located CL51 aerosol backscatter show exactly same variation for the PBL evolution except the higher backscatter between 12 to 14 EDT for CL51. That is because that cloud screen process was applied to the LMOL UVB aerosol retrieval process.

**5 Uncertainty**

The sensitivity of the algorithm to uncertainty in the input parameters is analyzed for August 28, 2018 case in this section. The aerosol extinction retrieval uncertainties caused by the lidar detection noise, reference value estimation, atmospheric molecular density, ozone concentration uncertainty, and the $S_1$ will be discussed in this section. The quantitative estimation of the aerosol extinction and backscatter uncertainty is challenging, and no standardized recommendation exists (Leblanc et al., 2016b). In

this work, the total uncertainty of the retrieved extinction coefficient is calculated by following standard propagation of error practices. The retrieved aerosol profile depends on several instrumental and physical parameters for the lidar system. The measurement model for the system is presented as equation (9). The individual values y of the quantity Y was shown in equation (10) (Leblanc et al., 2016b).

$$Y = f(X_1, X_2, X_3, \ldots, X_N) \tag{9}$$

$$y = f(x_1, x_2, x_3, \ldots, x_N) = y_0 + \sum_{n=1}^{N} \frac{\partial y}{\partial x_n} x_n \tag{10}$$

The combined standard uncertainty $u_y$ is obtained using the individual standard measurement uncertainties associated with the input quantities in the equation (9). As shown in equation (11), the combined standard uncertainty $u_y$ equals the positive squared root of the combined variance in case of all variables that are independent (Leblanc et al., 2016b).

$$u_y^2 = \sum_{n=1}^{N} \left(\frac{\partial y}{\partial x_n}\right)^2 u_n^2 \tag{11}$$

As shown in section 2, the signal was used to calculate the aerosol extinction noted as $X(R)$ and shown as equation (12).



$$X(R) = [P(R) - P_0]R^2 \exp\left[2 \int_0^R \sigma_{O_3} N_{O_3}(r)\, dr\right] \tag{12}$$

The detection noise uncertainty is derived from Poisson statistics associated with probability of detection of a repeated random event. Following Leblanc et al., 2016b, the subscript (DET) was used for detection noise. The uncertainty in the raw signal $P(R)$ caused by detection noise could be expressed as equation (13) and reflect purely random effects during detection (Russell et al., 1979).

$$u_{P(DET)}(R) = \sqrt{P(R)} \tag{13}$$

It is propagated to the background and $O_3$ corrected signal $X(R)$ by apply equation (11) to equation (12):

$$u_{X(DET)}(R) = R^2 \exp\left[2 \int_0^R \sigma_{O_3} N_{O_3}(r)\, dr\right]\sqrt{P(R)} \tag{14}$$

This propagated to the retrieved aerosol extinction $\alpha_1$ by apply equation (11) to equation (4):

$$u_{\alpha_1(DET)}(R) = \frac{\partial \alpha_1(R)}{\partial X(R)}\, u_{X(DET)}(R) \tag{15}$$

The $O_3$ uncertainty is noted as $u_{o_3}$, and it is propagated to the background and $O_3$ corrected signal $X(R)$ by apply equation (11) to equation (12):

$$u_{X(O_3)}(R) = \frac{\partial X(R)}{\partial O_3(R)}\, u_{o_3} \tag{16}$$

This propagated to the retrieved aerosol extinction $\alpha_1$ by apply equation (11) to equation (4):

$$u_{\alpha_1(O_3)}(R) = \frac{\partial \alpha_1(R)}{\partial X(R)}\, u_{X(O_3)}(R) \tag{17}$$

This propagated to the retrieved aerosol extinction $\alpha_1$ by apply equation (11) to equation (4):

$$u_{\alpha_1}(R) = \sqrt{\begin{array}{l}\left(\frac{\partial \alpha_1(R)}{\partial X(R)} u_{\alpha_1(DET)}(R)\right)^2 + \left(\frac{\partial \alpha_1(R)}{\partial X(R)} u_{\alpha_1(O_3)}(R)\right)^2 + \left(\frac{\partial \alpha_1(R)}{\partial \beta_2(R)} u_{\alpha_1(N_m)}(R)\right)^2 \\ + \left(\frac{\partial \alpha_1(R)}{\partial S_1(R)} u_{\alpha_1(S_1)}(R)\right)^2 + \left(\frac{\partial \alpha_1(R)}{\partial \alpha_1(R_c)} u_{\alpha_1\left(\alpha_{1(R_c)}\right)}(R)\right)^2\end{array}} \tag{18}$$

The uncertainty shows in equation (18) take into account the impact of integral uncertainty from targeted altitude to the reference point because the equation (4) has the integral taking account the molecular extinction and the $O_3$ corrected return lidar signal of the target altitude to the reference point. The $u_{\alpha_1(N_m)}(R)$ is the atmospheric molecular number density uncertainty we use as 1% following the result from Kuang et al., 2020. The $S_1$ is assigned $35 \pm 15$ for this example and the uncertainty of the $u_{\alpha_1(S_1)}(R)$ is about $\pm 40\%$. The uncertainty for the reference value is taken as 10 times for the total uncertainty analysis as show in equation (18).



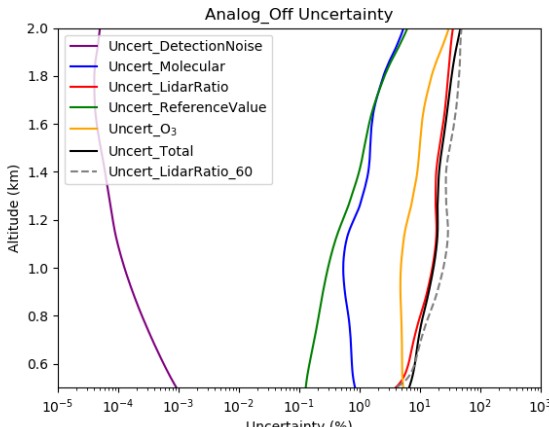
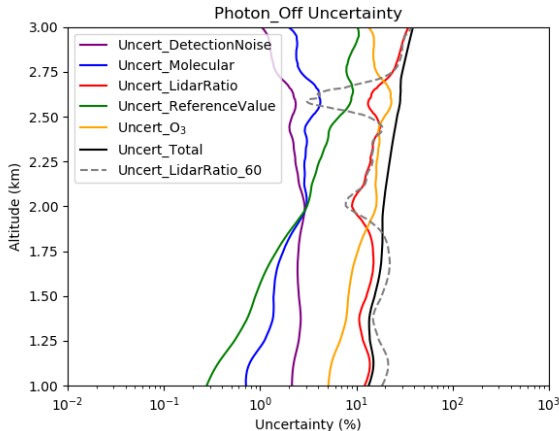

Figure 7: The uncertainty budget for the LMOL Analog channel (left) and the Photon channel (right) for August 28, 2018 afternoon retrieval. The uncertainties are attributed to different factors: detection noise (purple), molecular number density (blue), lidar ratio (red), reference value (green), uncertainty of $O_3$ (orange), total uncertainty (blue). The uncertainty caused by using 60 sr as $S_1$ was shown in dashed gray line.

We calculate uncertainties of the analog channel and the photon channel separately, so we could assess how the different parameters impact the retrieval uncertainty for both channels. As figure 7 shown, the uncertainty caused by detection noise is very small for both channels. The uncertainty caused by the reference value and the molecular are less than 10% for both channels. Ozone uncertainty are 10% for the LMOL system and cause mostly less than 20% uncertainty for the analog channel and photon channel. The uncertainty of the $S_1$ cause about 4%-30% uncertainty for both analog and photon channel. The uncertainty of the $S_1$ and $O_3$ dominate the total uncertainty for both channels. We also show the uncertainty caused by using 60 sr as $S_1$ for UVB aerosol retrieval for afternoon August 28, 2018. It shows that $S_1$ equals 60 will increase aerosol retrieval uncertainty in PBL but uncertainty didn't change much above 2 km except in a layer located at 2.6 km.

## 6 Conclusion

For the first time, the aerosol extinction coefficient profile was retrieved from the LMOL 292nm attenuated backscatter using the Fernald algorithm are compared with airborne HALO data. A partial AOD difference method was introduced to determine the optimized value for 292 nm $S_1$ and AE between 292 nm and 532 nm which will be used for the LMOL 292 nm aerosol extinction retrieval. This optimized $S_1$ values for 292nm, and AE (292 & 532nm) improve the accuracy of the UVB aerosol retrieval compared with a prior estimation of these parameters. Furthermore, both 292nm $S_1$ and the AE (532 & 292 nm) can improve our understanding of the UVB optical properties of the transported Canadian smoke. The inter-comparison between HALO observations and 292nm LMOL TOLNet ozone lidar aerosol product was applied for example case on August 28, 2018, and result show very good agreement after the optimization method was applied. The retrieved LMOL 292 nm aerosol was also compared with co-located ceilometer and CCNY aerosol lidar close to LMOL system. These comparison shows that

LMOL UVB aerosol retrieval could show very good aerosol variations and mixing layer evolution. Error analysis shows that the uncertainty from $O_3$ and $S_1$ dominate the 292nm LMOL aerosol retrieval and needs to be carefully considered in TOLNet retrievals of aerosol profiles. In cases when there is no HALO data, a-priori determinations from differing aerosol types based on this kind of analysis work will serve to provide reasonable $S_1$ at UVB wavelengths. Consequently, further research is needed to characterize $S_1$ and AE at UVB wavelengths.


Data availability. The LMOL $O_3$ raw data used in this study can be downloaded from the LMOL website: https://www-air.larc.nasa.gov/missions/TOLNet/. The LMOL $O_3$ used to correct the raw data can be download from the TOLnet website: https://www-air.larc.nasa.gov/cgi-bin/ArcView.1/TOLNet?NASA-LARC=1#0. The HALO aerosol backscatter/extinction data used in this work can be downloaded from the LISTOS website: https://www-air.larc.nasa.gov/cgi-bin/ArcView/listos.


**Author contributions**

LL and TB formulated the overarching research goals. GG supported the $O_3$ data analysis which is a key factor for the aerosol retrieval algorithm. LL and JS did the UV aerosol extinction retrieval calculation. LL did lidar ratio and AE determination, aerosol extinction comparison between LMOL and HALO. AN provide HALO aerosol data product and HALO instrument introduction in paper. YW and FM involve in CCNY lidar aerosol data and provide CCNY PBL height data. SK contributes to the UV aerosol retrieval algorithm. LL wrote the initial draft of the paper with contributions from all co-authors. All authors reviewed the manuscript.



**Competing interests**

The authors declare that they have no conflict of interest.

**Acknowledgement**

The authors gratefully acknowledge support by the NASA Postdoctural Program that enabled this study. LMOL and HALO lidar participation in the LISTOS campaign was made possible with funding by the NASA Tropospheric Composition Program. The LMOL team gratefully acknowledges the Connecticut Department of Energy and Environmental Protection for providing site support to enable LMOL operations at the Westport location. The HALO team acknowledges support from the Langley Research Center Research Services Division for the operation and maintenance of the King Air B200 Aircraft throughout the duration of the LISTOS campaign. We thank Susan Kooi and James Collins for their contribution to analysis and archival of the HALO HSRL dataset. We thank Daniel B. Phoenix for his valuable comments and his help in editing the







manuscript. Y.W. and F. M. are supported by the New York State Energy Resources Development Authority (grant # 137482), NESCAUM (grant # 2411) and NOAA-CESSRST under the Cooperative Agreement Grant # NA16SEC4810008.

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
