# Peer review of "Retrieval of UVB aerosol extinction profiles from the ground-based Langley Mobile Ozone Lidar (LMOL) system"

_Atmospheric Measurement Techniques, 2021_

## Author Comment (AC1)

***Reviewer's comments are in black***

***Answers to the reviewer's comment are in blue***

**Comments from reviewer for UV aerosol retrieval paper**

**The authors would like to thank the reviewer for you time and effort to help significantly improve the manuscript.**

**RC1**: 'Comment on amt-2021-307', Anonymous Referee #1

Thank you for expressing your concerns; we have significantly revised the abstract and introduction to improve clarity of the approach. We believe this will help to prevent any further misunderstanding.

This work present and iterative method to obtain lidar ratio at 292 nm, which is later applied, to Langley Mobile Ozone Lidar (LMOL) backscattered signal. Once lidar ratio at 292 is estimated, authors use the classical Klett method to obtain independent aerosol extinction and backscattering. It is well-known in lidar literature that Klett method cannot provide accurate estimates of extinction profiles because of possible variations of lidar ratio with height.

Nevertheless, the authors try to address an important challenge and provide an estimation of lidar ratio at 292 nm. Typically, backscattered lidars use co-located measurements of sun-photometry AOD for estimating lidar ratios (see for example MPLNET or EARLINET/ACTRIS retrievals). Currently, there are not many radiometric measurements that provide aerosol AOD at 292.

Answer: We use Fernald method for aerosol extinction retrieval. Instead of using the sun-photometer AOD to constrain the aerosol retrieval, the analysis in this manuscript uses the co-located HALO measurement to constrain the retrieval. The HALO provides aerosol backscatter, extinction, and lidar ratio profiles at 532 nm with high vertical resolution which is a more reliable constraint for the aerosol retrieval. This is because the sun-photometer is a column-only measurement (no profile) and also does not provide data at 292 nm. As a result, we use the iterative method to determine both the lidar ratio at 292 nm and AE between 292nm and 532 nm.

However, I do not rely in the approach presented by the authors. It might need further explanations. But as I understand they propose iterative variations of lidar ratio in LMOL system and provide different aerosol extinction profiles. The range of variation of lidar ratios is not enough because absorbing aerosol can present lidar ratios larger than 90, and OMI satellite retrievals demonstrated the importance that aerosol absorption might have extinction. On the other hand, I understand that they vary Angström exponent iteratively in HALO system to obtain equivalent aerosol extinction at 292 nm. They are ignoring possible effects of variations of

Angström exponent with altitude. If so I would rely more on Angström exponent measurements
using sun photometry. Finally, for the evaluation they are using the same data that for the
computation in the iterative method, which is not appropriate.

Answer:  There are two reasons that we believe the range of variation of lidar ratio from 10 to
90 making sense. The first one is according to previous publication (Sasano and Nakane, 1984).
The second one is the result that we get from our calculation. Most 292 nm lidar ratio calculations
converge to values between 20 to 70 sr. That is why we selected the current lidar ratio range (10
sr –90 sr) to save calculation time.

The sun photometer cannot provide the variation of aerosol Angström exponent with altitude.
The sun photometer only provides the aerosol Angström exponent for the total column aerosol,
while LMOL lidar measures aerosol of lower part of troposphere. Aerosol Angström exponent
derived from our method using the LMOL data and HALO data at the same altitude range,
provides a more reliable result.  In addition, the sun photometer also cannot provide Angström
exponent between 292nm and 532 nm since there is no sun-photometer data available at
Westport site.

As stated in the paper, we are not proposing a new iterative variation based on some variation
of the lidar ratio. We are here comparing the LMOL retrieval, using several lidar ratio, to the HALO
data. HALO is an HSRL lidar, which means that it can retrieve the lidar ratio and Angström
exponent in an independent and reliable way (see refs cited). As a result, this represents the
state-of-the-art to provide aerosol parameters at each altitude.  In the present paper, we consider
the HALO value as the ground truth, and we compare it to the Fernald method applied to LMOL.
We have added an additional figure (Figure 1) and description at the beginning of the manuscript
to help clarify the analysis approach.

With all these points I propose to evaluate the method with CCNY lidar for 355 nm and make
intercomparisons with extinction coefficient at that wavelength computed by Raman
methodology.

Answer: The aerosol AE between 355 nm and 386.7 nm still need to be assumed when 355 nm
aerosol extinction was retrieved using Raman Lidar. HALO can independently obtain aerosol
extinction and backscatter using Rayleigh and Mie signal without any assumption. So, it is better
choice to evaluate our method with HALO data. Furthermore, quantitative CNNY comparisons to
LMOL retrievals, which may be interesting, are spatially too far away to be useful for this study.

Section 4.2 does not provide any relevant scientific results. It only shows coherence in the vertical
structures of aerosols, and for that it is not necessary to retrieve extinction coefficients.
Moreover, the study-case selected to demonstrate the novel methodology must be different
than that used for the validation.

Answer: The comparison of aerosol profile in section 4.2 is very important because it will show
the difference between LMOL and HALO aerosol profiles when you select the optimized lidar ratio and Angström exponent. This intercomparison is important because it illustrates the ability
of the LMOL aerosol retrieval to capture a consistent aerosol feature when compared to HALO
HSRL aerosol data. And thus, can produce relevant data for campaign analysis in the relationship
of aerosols to ozone features. It is important also because is a first study in the development of
a new data product for LMOL.

Finally, section 5 only shows that lidar ratio is the most important parameter in the retrieval of
backscattering and extinction, which is widely known in lidar community. What is necessary in
section 5 is sensitivity test of the new methodology proposed by the authors, which can be done
using synthetic data.

Answer: This section not only shows the lidar ratio being important, but uncertainties in $O_3$ are
comparable and change as a function of height. We agree lidar ratio is a key factor in the retrieval
of backscattering and extinction.  The sensitivity of both lidar ratio and angstrom exponent is
illustrated in section 3.2 and therefore believe an additional sensitivity analysis is not needed.

---

## Author Comment (AC2)

**\*\*\*Reviewer's comments are in black\*\*\***

**\*\*\*Answers to the reviewer's comment are in blue\*\*\***

**Comments from reviewer for UV aerosol retrieval paper**

**The authors would like to thank the anonymous reviewer for your time and effort to help significantly improve the manuscript.**

**RC2**: 'Comment on amt-2021-307', Anonymous Referee #2

Review of  "Retrieval of UVB aerosol extinction profiles from the ground-based Langley Mobile Ozone Lidar (LMOL) system" by Lei et al.,

This paper describes an algorithm for the aerosol extinction retrieval out of the Langley Mobile Ozone Lidar (LMOL) as compared to 20 coincident flights with the NASA Langley High Altitude Lidar Observatory (HALO) 532 nm aerosol extinction product. This work also accomplishes the first known 292nm aerosol product inter-comparison between HALO and Tropospheric Ozone Lidar Network (TOLNet) ozone lidar.

In general, this paper would benefit from an additional proofreading.

Major Comments:

In general, this is a very technically developed manuscript. Many of the equations are first principles and well known in the lidar community. In general, substitution of these for graphical elements (flow charts, or signal processing chains) would improve the readability. This also allows the author to highlight sections that are new to this original research.

Answer: We agree and added a diagram as Figure 1 and descriptive text at the beginning of the paper to better illustrate our approach and improve the readability.

The added flow chart and description of flow chart are as follows:

[Figure]

Figure 1. Flow chart for the approach used in this work.  The cyan section corresponds to the
processing needed for the retrieval of the optimal $(S_1 , AE)$
*"To retrieve the $S_1$  and AE, an iterative method with 3 main steps was used as shown in Figure 1.*
*The first step is the retrieval of the aerosol extinction at 292nm from LMOL. For that, the LMOL*
*raw data are corrected from the ozone absorption. Then the Fernald method (Fernald et al., 1972,*
*Fernald, 1984) is used with an empirical $S_1$  (which is modified in subsequent iterations to explore*
*the parameter space). For the current study, the impact of the aerosols was low enough that an*
*iterative correction to the $O_3$ density was not necessary to retrieve the aerosol extinction*
*accurately; for dense aerosols layers, the method described in Browell et al., 1985 would have*
*been used. The second step is the retrieval of the aerosol extinction at 292 nm from HALO. The*
*conversion of the extinction from 532nm to 292nm is done by using an assumed AE which is also*
*modified in subsequent iteration to explore the ($S_1$, AE) parameter space. The third step is the*
*comparison of the aerosol extinction from both instruments at 292 nm.  The integration of the*
*difference provides the partial aerosol optical depth (AOD) difference, refered later as the partial*
*AOD index. Once the plausible ($S_1$, AE) parameter space has been explored, there will be a*
*minimum to the partial AOD index which points to the best ($S_1$, AE) for the observed conditions.*
*The LMOL aerosol extinction profile related to optimized $S_1$ and difference between the LMOL and*
*HALO 292 nm aerosol profile related to the optimized $S_1$ and AE was also recorded for further*
*analysis."*
My major questions

Is this approach actually novel? The authors describe this as working from between 0.5 and 3.5km
– does this indicate it may only work properly or is biased for aloft/transported aerosol layers?
Please re-emphasize the importance of this work.

Answer:   ***a little more wordsmithing***

- Yes, this approach is unique because it provides a way to obtain lidar ratio at 292 nm and get the 292 nm aerosol retrieval for LMOL system. It also provides the AE between 292nm and the intercomparison between the LMOL and HALO system.
- This work focuses aerosol retrieval between 0.5 to 3.5 km because the restriction of the lidar measurement which is lower at daytime because of the strong background.
- Capturing aerosol extinction between 0.5 to 3.5 km is very useful because it will help us to retrieve the planetary boundary layer (PBL) height and also help us to learn aerosol property in the lower part of troposphere. Furthermore, aerosol profiling information can still play an important role for model intercomparisons and satellite retrievals.
- The extinction in wavelength less than 300 nm is difficult to be retrieved using simple lidar techniques because the impact of $O_3$ absorption and lacking information of lidar ratio at those wavelengths. We proposed the new method to retrieve aerosol extinction at 292 nm using Fernald method with combing the profile of ozone and the profile of 292-nm elastic backscattering from LMOL. The selection of lidar ratio is very import for the retrieval of aerosol extinction. However, the lidar ratios less than wavelength of 300 nm for different aerosol type are rarely discussed according to previous research.  So, HALO results were used to constrain LMOL retrieval to improve lidar ratio accuracy.
- Combing long-term measurements of HALO and LMOL, a database of lidar ratio for different aerosol type will be built and can improve LMOL aerosol extinction retrieval without relying on HALO measurements in future.

Can this method be extended to cases outside of when there were HALO overpasses? Otherwise
this does not have as much appeal to the general audiences.

Answer: As mentioned above, combing long-term measurements of HALO and LMOL, a database
of lidar ratio for different aerosol type can be developed < 300 nm and improve LMOL aerosol
extinction retrieval without HALO in future.

A missed opportunity is using the ceilometer to compare with – this is a 24/7 measurement that
is made very widely over the country. Then use the HALO data to act as a reference for the quality
of the results in this specialized case – and then improve confidence in the ceilometer derived
method.

Answer: Although it is not the focus of this work, we agree that applying similar HALO/ground-
lidar analyses to ceilometer networks could prove useful and worth pursuing as a separate study.
However, this is expected to be challenging due to the longer near-IR wavelength and related
signal-to-noise limitations of these systems.  As a result, we show a qualitative comparison with
the Westport ceilometer that is shown in Figure 6 (Figure 7 in new version of manuscript). With
respect to the overall approach described in this study, the ceilometer limits would not provide
quantitative benefit since it is not deriving the lidar ratio as a function of height and then requires
the a-priori assumptions that we are trying to avoid when comparing with LMOL.

The author described Canadian wildfire smoke, but I cannot tell clearly from any of the images
1) where the smoke resides, 2) how improved the retrieval is in these areas of smoke, or 3) what
effects the data set has made to improving the remote sensing of the optical properties of the
aerosols.

Answer:
(1) The measurement was impacted by the Canadian wildfire smoke and the active fire in
southeastern United States. As we mentioned in introduction: "The August 28 case was shown
in detail because the air quality exceedance during that day was probably caused by the impact
of long-range transport of wildfire emissions (Rogers et al., 2020)".  (We encourage the reviewer
to check figure 2 and figure 4 for Rogers et al., 2020 paper)
(2) For the wildfire emission case, some cases are directly has smoke layer aloft, but some cases
like the August 28 case the smoke is already mixed in the boundary layer and does not appear
as a distinct layer. We can tell that our measurement was impacted by smoke according to
previous publication (Wu et al., 2021; Rogers et al., 2020; Hung et al., 2020). The August LISTOS
lidar, HALO, and AERONET data provided guidance on which days smoke was expected to be
present.
(3) The optimized lidar ratio at 292 nm was smaller than lidar ratio at 532 nm, consistent with
expectations but also yields the most appropriate quantitative outcome for the aerosol
influenced by wildfire emission.

Minor Comments:

L55 – remove 'a lot'

Answer: We remove it.

L60 - Langley

Answer: We change it. Thank you for reminding us!

L64 "In this paper, the impact of the aerosols was low enough that an aerosol correction to the
O3 density was not necessary; otherwise, an interative process would have been necessary
(Browell et al., 1985)" – Suggest rephrasing this statement. This sounds like it basically voids the
need for this work. Although the ozone correction to the signals may not reduce accuracy, the
authors state the uncertainty in ozone is 10-20% - that must impact the uncertainty of the aerosol
correction.

Answer:  The objective of the present paper is to characterize the aerosols using the 292 data
from LMOL, not specifically to correct the $O_3$ from aerosol impact. Ultimately, when the technique is validated, this would be a step; however, the validation of the retrieval is partially in
the current paper. With the standard method for correcting the $O_3$ signal from the aerosol
contribution, (1) $O_3$ should be retrieved from the raw signal, which (2) allows to correct 292 nm
for (3) retrieving the aerosols extinction. From the aerosol, we could (4), correct the density of
$O_3$. With that, we could go back to (2) for correcting the signal. The procedure repeats until
convergence.

In the present paper, we have conditions that lead to a small correction of $O_3$ after the step (4),
which in turns gives a negligible change in the aerosols computed in the initial (3).

This is what we meant by "aerosol correction to the O3 density". We changed into "an iterative
correction to the $O_3$ density is not necessary to retrieve the aerosols accurately".

L66 – New paragraph for LISTOS

Answer: Yes, we give new paragraph for LISTOS instruction.

L105 - rather than 'raw' data, what is being analyzed? Range corrected-Elastic Backscatter
Profiles at 292?

Answer: The $O_3$ corrected range corrected-elastic backscatter profile at 292 nm was analyzed.

L132 – LISTOS

Answer: We change it. Thanks for reminding us!

2.4 The Ceilometer located nearby LMOL – consider moving this up to 2.2 since it was co-located
with 2.1.

Answer: That make sense. We move it.

L160 – Lidar ratio is introduced here but is frequently used in the text up until this point.

Answer: We define the lidar ratio at the first time we use it in the text (at line 51).

Section 3 could benefit from some sort of "flow chart" graphic. Or illustration of the changes in
the corrected signals after certain steps.

Answer: We add the flow chart in the introduction part (figure 1 in new version of manuscript)
which will be helpful to improve the readability of section 3.

Figure 1 caption - (a) August 28fternoon needs to be fixed. Are these derived vertically or for a
column?

Answer: Yes, we fix it as "Augst 28 afternoon ...". It is vertically average value derived from
HALO S1 profile.

Table 1- 521nm?

Answer: It is 532 nm. We changed it in new manuscript. Thank you for reminding us!

L280 – "These results show that by using the selected S1 and AE in Table 1, LMOL has the
capability to retrieve aerosol extinction in 292nm with reasonable accuracy.", What is the
estimated uncertainty of this retrieved aerosol component?

Answer: We could see the difference between the retrieved LMOL aerosol profile and the
converted HALO aerosol extinction are less than 10 % when use the optimized lidar ratio and
Angstrom exponent for August 28 afternoon case. The uncertainty of using lidar ratio other than
the optimized value for retrieval was show in figure 3 (b) (figure 4 (b) in the new version of
manuscript).

Figure 6 – Are you able to convert the ceilometer to 292nm? Please label in the plot title. Is the
PBL height detection "in-house" for the ceilometer or the Vaisala standard product?

Answer: We think it is hard to convert the ceilometer to 292 nm because the ceilometer only
provides the backscatter signal at 910 nm. We have added the wavelength information in the
plot title for each curtain plot in figure 6 ( figure 7 in new version of manuscript). The PBL height
is from the Vaisala standard product which could obtained from the LISTOS archive data. We
have added related information in section 4.2 in the manuscript.

Figure 7 - total uncertainty (blue) – should be black. Why is the analog Det Nois decreasing with
altitude? Wouldn't you expect as the signal to be much higher compared to background noise
values, that the uncertainty would increase? Is there a need to show both analog and photon
counting here?

Answer:

▪ Yes, the total uncertainty is black. We changed it in the caption.
▪ I think we need to point to equation 15 (equation 16 in the new version manuscript) here:
the detection noise is the coupling between the Udet, which increases with altitude and
the differential of the retrieved value with the detection rate, which could be decreasing
with altitude. We set up a value for the aerosols, fixes, at the higher altitude, then we go
down from there. The uncertainties are therefore adding while going downwards and are
considered stable at high altitude.
▪ We think it is better to show both analog and photon counting channel here because the
analog and photon counting are used for retrieval for different altitude ranges.

Figure 7  In some cases in Photon Counting it looks like the uncertainty in using 60sr is less than
using the technique  applied in this paper. Is that the case?

Answer:

In the sensitivity study, we apply the uncertainty algorithm to a specific lidar ratio and with an
uncertainty on that lidar ratio. If the retrieval equations were perfectly linear, the uncertainty of
the retrieval (in percent) would be proportional to the uncertainty of the lidar ratio (in percent)
and therefore constant with altitude. In that case, a lidar ratio of 60 +-40% or 35 +-40% would
lead to the same uncertainty in the retrieved aerosol extinction.

This is obviously not the case: a change of the lidar ratio leads to a change in both the retrieved
extinction and absorption, which could lead to uncertainties higher or lower depending upon the
density of aerosols and the amount of aerosols above.

Please note that in the sensitivity study, we assume the 40% uncertainty, i.e. without knowing
the optimized value of the lidar ratio which would fit better with the dedicated instrumentation.
The error is therefore composed both of the possible bias (poorly known ratio) and the noise due
to the change in altitude of the ratio.

Conclusions – rather than say 'good', please use specifics from the retrieval results.
Consequently, further research is needed to characterize S1 AE at UVB wavelengths – what
exactly is needed?

Answer: We add specific result in conclusion instead of using "good" or "very good". The new
statement are as follows:
*"The inter-comparison between HALO and LMOL aerosol products showed an agreement within*
*10% up to 3 km after the optimization method was applied in the case of August 28, 2018. The*
*retrieved LMOL 292 nm aerosol was also compared with co-located ceilometer and CCNY aerosol*
*lidar. It shows that LMOL could capture a consistent aerosol feature and mixing layer evolution.*
*Error analysis shows that the uncertainty from $O_3$ and $S_1$ dominate the 292 nm aerosol retrieval*
*and needs to be carefully considered in the retrievals of aerosol profiles of all the TOLNET Lidars.*
*In cases when there is no HALO data, a-priori determinations from differing aerosol types based*
*on this kind of analysis work will serve to provide reasonable $S_1$. Consequently, further research*
*is needed to characterize $S_1$ and AE at UVB wavelengths: first, an effort should be made on*
*determining the variation of $S_1$ and AE with altitude by carefully addressing the uncertainties in*
*the HALO $S_1$ profile products; second, additional co-located LMOL/HSRL measurements should be*
*done to evaluate $S_1$ and AE for different aerosol types (smoke, dust, marine aerosol, and pollutant*
*aerosol). This characterization could ultimately enable the use of equipment with a better*
*availability than an HSRL (examples of such equipment could be the MPLs) to provide the ancillary*
*data necessary for the aerosol extinction retrieval."*

---

## Author Comment (AC3)

**We have rewritten the abstract as following:**

"*Abstract. Aerosols emitted from wildfires are becoming one of the main sources of poor air quality in the US mainland. Their extinction in UVB (wavelength range 280-315 nm) is difficult to be retrieved using simple lidar techniques because of the impact of $O_3$ absorption and the lack of data about the lidar ratios at those wavelengths. Improving the characterization at these wavelengths will enable their monitoring with different instruments and also will permit to correct the aerosol impact on the ozone lidar data. The 2018 Long Island Sound Tropospheric Ozone Study (LISTOS) campaign in the New York City region brought a comprehensive set of instruments that enabled the characterization of lidar ratio for UVB aerosol retrieval. The NASA Langley High Altitude Lidar Observatory (HALO) produced the 532 nm aerosol extinction product along with the lidar ratio for this wavelength by using a high spectral resolution technique. The Langley Mobile Ozone Lidar (LMOL) is able to compute the extinction provided it has the lidar ratio at 292nm. The lidar ratio at 292nm and the Ångström Exponent (AE) between 292 nm and 532nm for the aerosols were retrieved by comparing the two observations using an optimization technique. We evaluate the aerosol extinction error due to the selection of these parameters, usually done empirically for 292nm lasers. This is the first known 292nm aerosol product inter-comparison between HALO and Tropospheric Ozone Lidar Network (TOLNet) ozone lidar. It also provided the characterization of the UVB optical properties of aerosol in the lower troposphere affected by transported wildfire emission.*"

The revised part in introduction (added flow chart and description of flow chart) are as follows:
"*To retrieve the $S_1$ and AE, an iterative method with 3 main steps was used as shown in Figure 1. The first step is the retrieval of the aerosol extinction at 292nm from LMOL. For that, the LMOL raw data are corrected from the ozone absorption. Then the Fernald method (Fernald et al., 1972, Fernald, 1984) is used with an empirical $S_1$ (which is modified in subsequent iterations to explore the parameter space). For the current study, the impact of the aerosols was low enough that an iterative correction to the $O_3$ density was not necessary to retrieve the aerosol extinction accurately; for dense aerosols layers, the method described in Browell et al., 1985 would have been used. The second step is the retrieval of the aerosol extinction at 292 nm from HALO. The conversion of the extinction from 532nm to 292nm is done by using an assumed AE which is also*

*modified in subsequent iteration to explore the (S₁ , AE) parameter space. The third step is the comparison of the aerosol extinction from both instruments at 292 nm. The integration of the difference provides the partial aerosol optical depth (AOD) difference, refered later as the partial AOD index. Once the plausible (S₁ , AE) parameter space has been explored, there will be a minimum to the partial AOD index which points to the best (S₁ , AE) for the observed conditions. The LMOL aerosol extinction profile related to optimized S₁ and difference between the LMOL and HALO 292 nm aerosol profile related to the optimized S₁ and AE was also recorded for further analysis."*

[Figure]

*Figure 1. Flow chart for the approach used in this work. The cyan section corresponds to the processing needed for the retrieval of the optimal (S₁ , AE)*

---

## Author Response (AR2)

**Comments to the author**: (from editor)

Although the paper has been improved a few additional details, as indicated below, require attention:

The authors would like to thank the reviewers and editor for your time and effort to help us improve the manuscript.

1. Figure 1 and its discussion do not belong in the introduction.

Answer: Thanks for your comments! We move the figure 1 and discussion of the flow chart to section 3. And add some explanation in the instruction part.

2. The uncertainty related to analog detection/APD is not as simple as using the equations derived for random errors due to shot noise in PMTs (eq.15/16). These are assumed to be a Poisson distribution, indicating that the arrival at the photodetector of photons for these signals is a Poisson stochastic process. For Poisson-distributed signals, a proportional, one-to-one relationship is known to exist between the mean of a distribution and its variance. I expect that the photocurrent from the APD no longer follows this string Poisson distribution, but there still may be a quantitative estimate of this uncertainty. In the example provided in the current manuscript, at a given altitude (say 1km) does not make physical sense to have an uncertainty that is 3-4 orders of magnitude lower for an APD than a PMT. There has to be some additional care in this and related to Fig 8. Is there a bias or electronic offset that needs to be accounted here?

Answer: We thank the reviewer for this insight on the error in this part. Technically speaking, we are following the Leblanc et al. papers that separate the uncertainty from the shot noise and the uncertainty arising from the analog-to-digital conversion, which means that our figures are still valid, but lack that specific noise uncertainty. We recalculate the detection noise according to the method mentioned in Liu 2006 et al, and figured out that the detection noise uncertainty for analog channel (as show in the new version of figure 8 (left)).

[Figure]

[Figure]

Figure 8: The uncertainty budget for the LMOL Analog channel (left) and the Photon channel (right) for August 28, 2018 afternoon retrieval. The uncertainties are attributed to different factors: detection noise (purple), molecular number density (blue), $S_1$ (red), reference value (green), uncertainty of $O_3$ (orange), total uncertainty (black). The uncertainty caused by using 60 sr as $S_1$ was shown in dashed gray line.

3. Is 90 sr a reasonable upper limit lidar ratio value in the UV? Please include a reference or state if confirmed by the authors own calculations.

Answer: We agree that the lidar ratio could exceed 90 sr in UV in some extreme cases. But we believe 90 sr is a reasonable upper limit lidar ratio value in our study. Prior research indicates the lidar ratio less than 90 for 532 nm for different aerosol type (Omar et al., 2009; Müller et al., 2005). And lidar ratio in UV wavelength is smaller than that in visible 532 nm for aged smoke particle (Haarig et al., 2018, Müller et al., 2005; Müller et al., 2007; Ortiz-Amezcua., 2017). This is the key reason why we believe 90 sr is a reasonable upper limit. In addition, we can see that calculation converge between 20 and 70 sr as shown in figure 4 (a). That is why we selected the current lidar ratio range (10 sr –90 sr) to save calculation time.

Reference:

Haarig, M., Ansmann, A., Baars, H., Jimenez, C., Veselovskii, I., Engelmann, R., and Althausen, D.: Depolarization and lidar ratios at 355, 532, and 1064 nm and microphysical properties of aged tropospheric and stratospheric Canadian wildfire smoke, Atmos. Chem. Phys., 18, 11847–11861, https://doi.org/10.5194/acp-18-11847-2018, 2018.

Liu, Z., Hunt, W., Vaughan, M., Hostetler, C., McGill, M., Powell, K.,Winker, D., and Hu, Y.: Estimating random errors due to shot noise in backscatter lidar observations, Appl. Optics, 45, 4437– 4447, 2006.

Müller, D., Mattis, I., Wandinger, U., Ansmann, A., Al- thausen, D., and Stohl, A.: Raman lidar observations of aged Siberian and Canadian forest fire smoke in the free troposphere over Germany in 2003: microphysical particle characterization, J. Geophys. Res., 110, D17201, https://doi.org/10.1029/2004JD005756, 2005.

Müller, D., Ansmann, A., Mattis, I., Tesche, M., Wandinger, U., Althausen, D. and Pisani, G.: Aerosol-type-dependent lidar ratios observed with Raman lidar, J. Geophys. Res., 112, D16202, https://doi.org/10.1029/2006JD008292, 2007.

Omar, A. H., Winker, D. M., Vaughan, M. A., Hu, Y., Trepte, C. R., Ferrare, R. A., Lee, K. P., Hostetler, C. A., Kittaka, C., Rogers, R. R., and Kuehn, R. E.: The CALIPSO Automated Aerosol Classification and Lidar Ratio Selection Algorithm, J. Atmos. Ocean. Tech., 26, 1994–2014, https://doi.org/10.1175/2009JTECHA1231.1, 2009.

Ortiz-Amezcua, P., Guerrero-Rascado, J. L., Granados-Muñoz, M. J., Benavent-Oltra, J. A., Böckmann, C., Samaras, S., Stach- lewska, I. S., Janicka, L., Baars, H., Bohlmann, S., and Alados-Arboledas, L.: Microphysical characterization of long-range transported biomass burning particles from North America at three EARLINET stations, Atmos. Chem. Phys., 17, 5931–5946, https://doi.org/10.5194/acp-17-5931-2017, 2017.